

# Comparison of electrical collection topologies for multi-rotor wind turbines

Paul Pirrie[1], David Campos-Gaona[1], and Olimpo Anaya-Lara[1]

[1]Wind and Marine Energy Systems CDT, University of Strathclyde, Royal College Building, 204 George Street, Glasgow, Scotland

**Correspondence:** Paul Pirrie (paul.pirrie@strath.ac.uk)

**Abstract.** Multi-rotor wind turbines (MRWT) have been suggested in literature as a solution to achieving wind turbine systems with capacities greater than 10 MW. MWRT's utilise a large number of small rotors connected to one support structure instead of one large rotor, with the aim of circumventing the square cube law. Potential benefits of MRWT's include cost and material savings, standardisation of parts, increased control possibilities and improved logistics for assembly and maintenance. Almost

all previous work has focused on mechanical and aerodynamic feasibility, with almost no attention being paid to the electrical systems. In this research eight different topologies of the electrical collection network for MRWT's are analysed to assess which are the most economically and practically viable options. AC and DC collection networks are presented in radial, star, cluster and DC series topologies. Mass, capital cost and losses are estimated based on scaling relationships from academic literature and up to date commercial data. The focus of this study is the assessment of the type of electrical collector topology

so component type and voltage level are kept consistent between topology designs in order to facilitate a fair comparison. Topologies are compared in terms of four main criteria; capital cost, cost effectiveness, total mass, and reliability. The most suitable collection topology for MRWT's is shown to be of the star type, in which each turbine is connected to the step up transformer via its own cable. DC topologies are generally found to be more expensive when compared to their AC counterparts due to the high cost of DC-DC converters and DC switchgear.

**1 Introduction**

As the wind industry tries to continue reducing the cost of energy, it is desirable to have as much rated capacity on one support structure as possible, particularly offshore. The substructure of a wind turbine is typically a large portion of the capital cost so increasing the size and power rating of the turbines is an obvious way to reduce the cost of energy (Manwell et al., 2014). Additionally, there is less sites to be maintained, which can drastically reduce the O&M costs of a wind farm, particularly in

offshore environments when access can be difficult. This has led the industry to develop very large wind turbines like the GE Haliade-X, with a rotor diameter of 220 m and a power rating of 12 MW. However, there is physical and economical limits as to how large single rotor wind turbines can become. It is shown in (Sieros et al., 2012) that as the radius of a wind turbine rotor increases, the loads due to self weight that are encountered in the blades and tower increase at a faster rate. This implies that more material and hence higher capital costs are required to manufacture blades and towers as rotors increase in diameter.





There are additional practical limits as to how large wind turbine components can become. For onshore wind turbines, the size
of components has already reached a practical limit due to difficulties involved in transportation of large components. Larger
components are possible when installed offshore, but there is still significant issues involved with transporting and installing
large components. Large components require large vessels to transport and install which come at a significant cost to installers.

Multi-rotor wind turbines (MRWT) offer an alternative solution to achieving wind turbine systems with large scale power
capacity. The idea is to have a large number of small turbines on one support structure instead of one very large rotor, circum-
venting the square cube law and achieving significant savings on material costs for blades and drive train components. It is
shown in (Jamieson and Branney, 2012) that the blades and major drive train components in a MRWT have $1/\sqrt{n}$ times the
mass of an equivalent single rotor system where $n$ is the number of rotors, implying significant savings in material costs for
MRWT's.

Various studies within the literature have shown that the MRWT concept has potential and should be investigated further.
An initial investigation of support structures required for MRWT's is conducted in (Manwell et al., 2014). Although the total
mass of the MRWT is shown to be higher than that of an equivalently sized single rotor turbine, the cost of the MRWT is
approximately 22% lower. This is due to a reduction of expensive materials required for blade and drive train components.
Various loading scenarios were investigated in the Innwind study (Jamieson et al., 2015). It was shown that MRWT's benefit
from reduced loading in all scenarios compared to that of an equivalently sized single rotor. An averaging effect is also demon-
strated, which leads to a smoother load profile over the MRWT structure, which could result in increased fatigue life. Various
studies performed at Kyushu University in Japan (Göltenbott et al., 2017; Ohya et al., 2017; Goeltenbott et al., 2015) have
shown that clustering turbines together can improve their performance. All of these studies show that clustering conventional
turbines results in modest gains in power, whereas clustering wind-lens turbines can produce significant increase in power.
This was also shown through simulations in the Innwind project, where a MRWT consisting of 45 turbines is expected to have
an increased power coefficient of 8%. Operation, maintenance and installation costs are also expected to be reduced using
the MRWT concept, as components would be small enough to use small vessels without specialist equipment required. It is
proposed in the Innwind project that each MWRT platform would be equipped with an on-platform crane capable of removing
and replacing an entire rotor assembly with no additional lifting equipment required. MRWT's would also benefit significantly
from an increased redundancy. If one small rotor fails only a small portion of the total power is lost and can be replaced at
the next regular service interval without significant loss of revenue. The Innwind project also compares the levelised cost of
energy (LCOE) of the conceptual 20 MW 45 rotor MRWT with two 10 MW reference wind turbines (RWT) and estimates that
the MRWT would have a reduction in LCOE of 15%. Some disadvantages of the MRWT concept are more complex support
structure design and increased component count.

Although some work has been done to provide insight into the feasibility of MRWT's, the electrical system has been largely
neglected. The aim of this study is to assess various electrical collection topologies that could be used in a MRWT in order
to determine the most suitable option. There are several areas that must be considered when identifying possible electrical
collection topologies for MRWT's. The cost of the system should be minimised as to not outweigh the savings in material
costs. Mass distribution must be optimised in order to avoid unnecessary reinforcement of the support structure. The ability to



vary rotational speed of individual rotors is an important aspect of power maximisation and load alleviation, so the independent operation of individual rotors is an important consideration. It is desirable to maximise the built in redundancy within a MRWT system, so reliability of components and redundancy within the electrical system is an important consideration.

This study performs an initial analysis of eight different electrical collection topologies that could be used within a 45 rotor MRWT system. Section 2 deescibes the methodology used, Section 3 describes the system outline, Section 4 discusses the
design constraints of the system, Section 5 outlines the proposed electrical collection topologies, Section 6 describes how the mass, cost and losses of each component has been estimated, Section 7 desribes how the cost effectiveness of each system is compared, Section 8 presents the results of the analysis and conclusions are made in Section 9.

## 2  Methodology

In this initial analysis of different electrical systems, the emphasis is placed on determining the most suitable *type* of system
so no detailed design work for each component will be conducted within this study. Instead, various AC and DC electrical topologies are designed using realistic components that are trusted and understood by the wind industry. If the benefits of the MRWT concept can be shown to be true while still using known and well understood components, it is more likely to be accepted as a viable alternative to the single rotor wind turbine. Mass, cost and losses are estimated for each component using information from industry data sheets where possible.

A number of steps were carried out in this study:

  – Design constraints were established.

  – Eight collection topologies were designed. Both AC and DC topologies are included. The focus of the study is to analyse the type of collection topology, so component type, voltage level and number of turbines in strings and clusters is kept consistent across all designs where possible.

– Capital cost and mass of each collection topology were estimated using a combination of scaling relationships and up to date commercial information.

  – Loss profiles of each component were used to estimate the total losses of each collection topology over the entire operating range of wind speeds.

  – Cost effectiveness of each collection topology was assessed based on the capital cost and total losses over a 20 year
lifetime of the project.

## 3  System outline

A MRWT system consisting of 45 rotors will be considered in this study. Figure 1 shows the physical layout of such a system with each circle representing a single wind turbine rotor. This number of rotors allows a balanced and compact design and



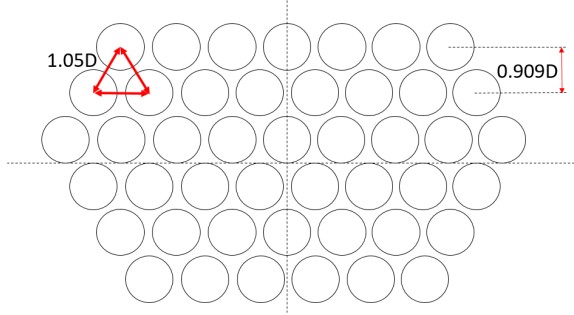

**Figure 1.** Proposed layout of a 45 rotor MRWT system.

is used in various studies such as (Jamieson and Branney, 2012) and (Jamieson et al., 2015). The Innwind study provides a

conceptual design for such a system and also makes comparisons to an equivalent system consisting of two 10MW RWT's from DTU to highlight the reduction in LCOE that could be realised by the MRWT concept. It is therefore desirable to use a similar system in this study to allow for easy comparison to both systems. Each rotor has a diameter of 41 $m$ which results in the same total swept area as the two 10 MW RWT's. Using examples of commercial wind turbines of this approximate size of rotor, a representative rated power of 500 kW was selected for each turbine, giving the system a total rated power of 22.5 MW.

## 4  Design Constraints


The following design constraints have been used to design the electrical topologies:

1. The system in question must have 45 rotors: to facilitate fair comparison with results in the Innwind project.

2. Each turbine must have a rating of 500 $kW$, diameter or 41 $m$, rated wind speed of 11.5 $m/s$, maximum coefficient of power ($C_P$) of 0.45, and rated rotor speed of 30 $rpm$: these values have been selected as representative values from

available wind turbines of this scale used within the industry.

3. Each turbine must have independent speed control: This is required in order to maximise energy capture and minimise loading on blades and drive train components.

4. It is assumed that each turbine operates in a maximum $C_P$ tracking mode below rated wind speed, and then power is held constant at rated power above rated wind speed via pitch control.


5. Each topology must connect to an AC collection network at $33kV$: this is a common voltage level used within the wind industry for collection networks. DC collection networks have been discussed in literature, but there is still no real world applications.





6. It is assumed that there is a platform at the top of the tower that is large enough to contain a transformer and/or converter for the multi-rotor array.

## 5  Proposed Electrical Topology Designs

A total of eight topologies were designed and each are described in this section. Each topology is based on commonly discussed collection network topologies for offshore wind farms.

The AC star and DC star topologies are shown in Fig. 2 and Fig. 3 respectively. In both the AC and DC star topologies each turbine is connected via its own cable to the converter or transformer situated at the base of the structure. The AC star uses a 3.3kV permanent magnet synchronous generator (PMSG) with a fully rated back-to-back IGBT based voltage source converter (VSC). The DC star uses a 1.5 kV PMSG with controlled IGBT based VSC. Various types of DC output wind turbines have been suggested in the literature with some utilising diode rectifiers to reduce weight and cost, and others utilising controlled IGBT based rectifiers. The advantage of using an IGBT based VSC is that it is able to easily control both the torque of the generator as well as the reactive power to the generator. This allows for independent speed control of each turbine while maintaining a constant DC output voltage, and also allows the use of any type of generator. A controlled rectifier has been selected for these reasons for each DC topology in this study. To allow for a fair comparison between topologies, all use a PMSG. The DC star topology uses a lower rated generator in order to have the DC cable voltage directly comparable to the AC star topology, which facilitates a fair comparison between the two and keeps the emphasis on the type of system rather than system voltage. Medium voltage (MV) generators are required to avoid high conduction losses in low voltage (LV) cables and also to remove the need for a transformer within the nacelle. The star topologies benefit from excellent redundancy as a fault in any one component within the array only results in the loss of 1/45th the total rated power. The disadvantage is that there is a large total cable distance, but cables with a smaller cross sectional area (CSA) can be used compared to other topologies. The DC star topology also may benefit from greater efficiency and lower mass due to fewer conversion steps and the use of DC cables (DC cables are known to be lighter and smaller compared to AC cables (Lakshmanan et al., 2015)).

The AC and DC cluster topologies are shown in Fig. 4 and Fig. 5 respectively. The cluster topologies gather power from a number of rotors, step up the voltage using either an AC transformer or a DC-DC converter and then transmit the power to the converter/transformer at the base of the structure. This allows for the use of industry standard 690 V generators as the cable distance between each rotor and the transformer or DC-DC converter is very small. Cluster topologies use much smaller cable distances compared to the star topologies, but require cables with larger CSA to handle higher currents. DC-DC converters are used in the DC topology and a 50/60 Hz transformer is used in the AC topology. DC-DC converters are smaller and lighter compared to AC transformers (Lakshmanan et al., 2015) but are also significantly more expensive, so it is expected that the DC cluster topology be more expensive but lighter than the AC cluster topology. The main disadvantage of the cluster topologies is the failure of the cluster transformer/DC-DC converter would result in the loss of the entire cluster of turbines.

The AC and DC radial topologies are shown in Fig. 6 and Fig. 7 respectively. These are based on the most common type of offshore collection network; the AC radial collection network (Bahirat et al., 2012). As it is the most common configuration,



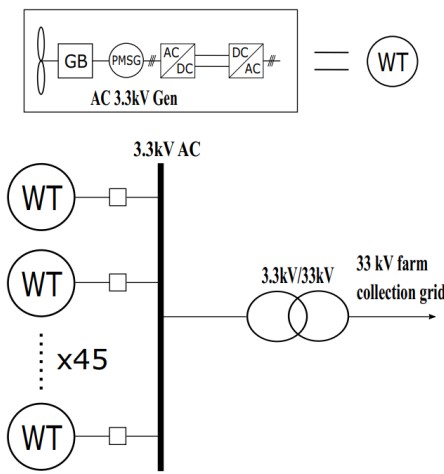

**Figure 2.** AC star topology.

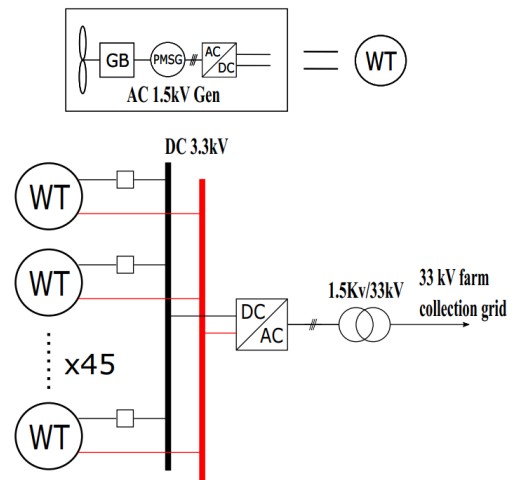

**Figure 3.** DC star topology.

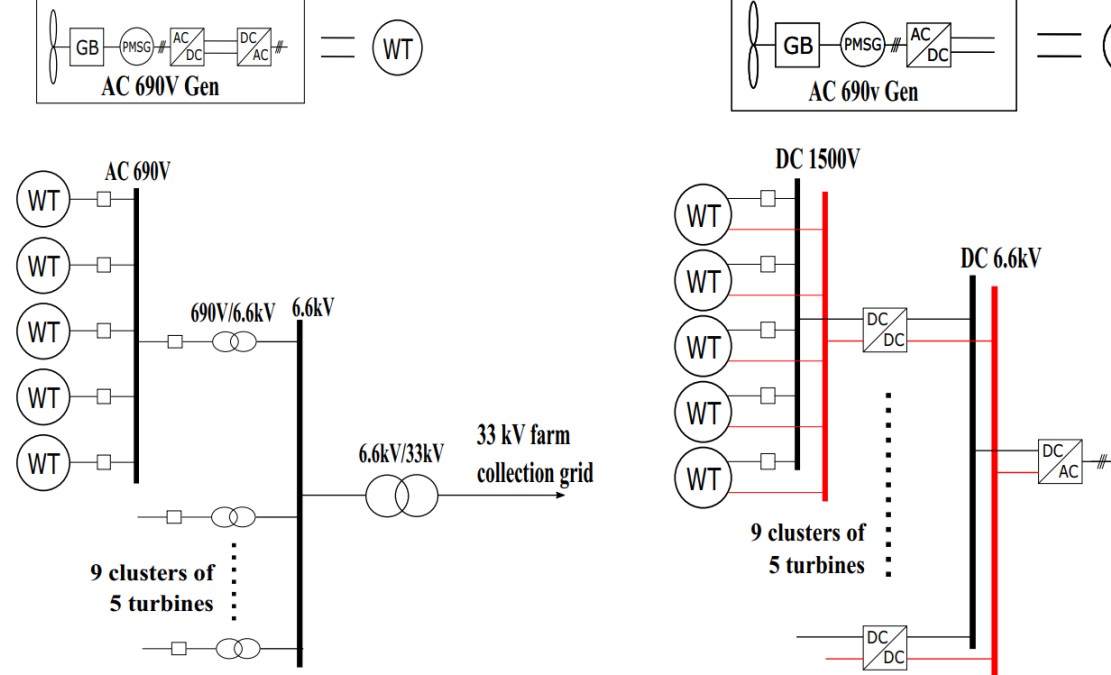

**Figure 4.** AC cluster topology.

**Figure 5.** DC cluster topology.

the AC radial topology will be used as the base topology throughout this study. In the radial topologies, a number of turbines are connected to a feeder cable which transmits the power to the transformer/converter at the base of the support structure. The





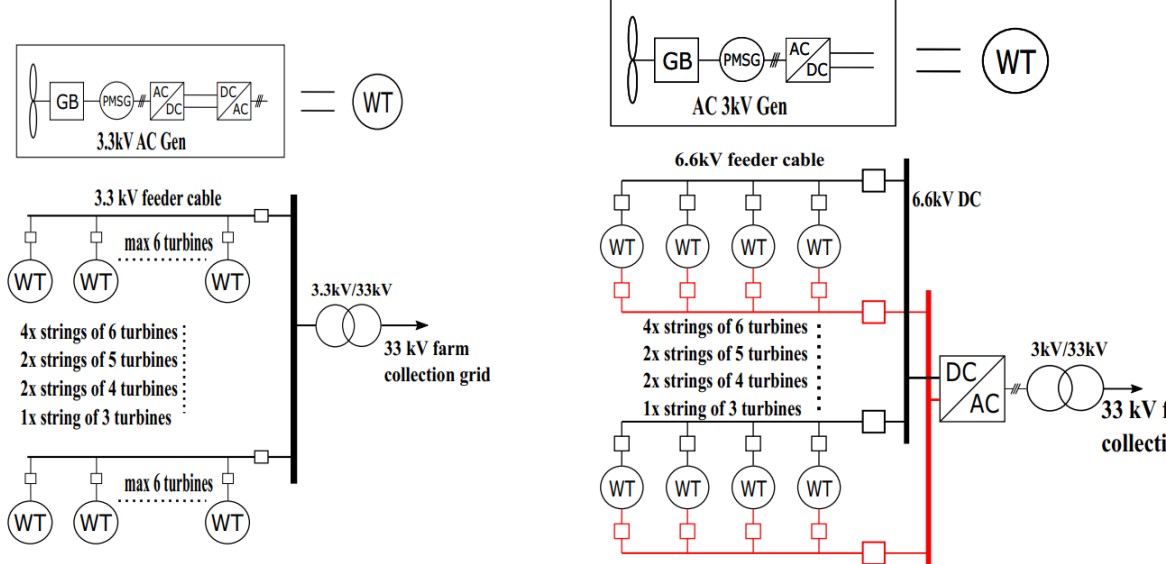

**Figure 6.** AC radial topology.

**Figure 7.** DC radial topology.

amount of turbines connected to a feeder cable is determined by the current carrying capacity of the cable and the power output of the turbines. For simplicity, both AC and DC radial topologies use non tapered cables in each string. Use of a 3.3kV feeder

cable in the DC radial design was originally used, but after initial assessment it was deemed unrealistic due to the large CSA of cables required so the next standard voltage level of 6.6 kV was selected. Radial designs benefit from short cable distances, simple design and operational experience. The main drawback is poor reliability as a fault in a feeder cable would result in the loss of the entire string. It should be noted that failure rates for cables are significantly lower than that of power electronic converters or generators.

DC series and DC series/parallel connected wind farm collection networks have been discussed in the literature and show enough promise to be included in this study (Bahirat et al., 2012; Ng and Ran, 2016; Lundberg, 2003). The main idea behind the DC series topology is to connect DC output turbines together in series to increase the voltage of the string without the use of AC transformers or large DC-DC converters, resulting in a very lightweight system. Figure 8 shows the DC series topology and Fig. 9 shows the DC series/parallel topology. In the DC series topology, a standard 690V AC PMSG is used with a controlled

rectifier to produce a DC output of 1.5 kV. Each string contains five turbines connected in series to produce a string voltage of 7.5 kV. Generator torque control is performed by the controlled rectifier and the string DC-DC converter maintains the DC voltage of the string. A fault in one turbine can be isolated by using a circuit breaker that operates in short circuit, maintaining a path for the DC current within the string. In the case of a fault within the string, the DC-DC converter at the end of the sting can vary its duty ratio to maintain the 11kV bus voltage. This system should have a low mass due to the small number of converters

used, but may be expensive due to the use of DC-DC converters and more expensive DC protection devices. There may also



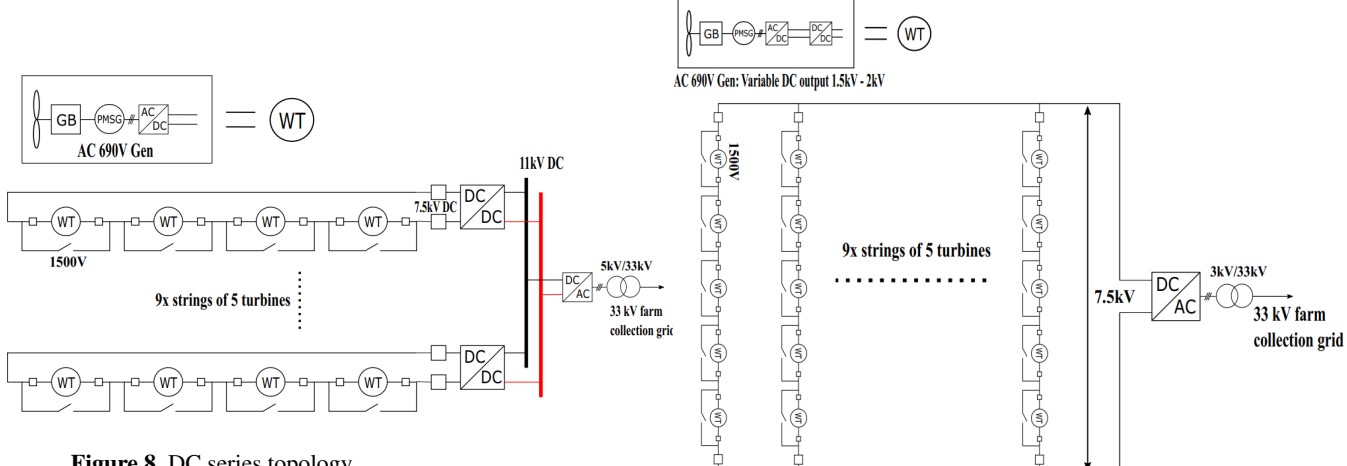

**Figure 8.** DC series topology.

**Figure 9.** DC series/parallel.

be issues regarding insulation as some components will require an insulation level high enough to withstand the whole string voltage to ground. This will be explored further if the topology shows promise.

The DC series/parallel topology is similar to the DC series topology in that is utilises DC output turbines connected in series to produce a high voltage within strings. It utilises a variable voltage output DC wind turbine where the generator is connected to a controlled rectifier and a DC-DC converter. As multiple strings of DC series turbines are connected together in parallel, the voltage of each sting must be kept the same as the others. This is achieved by the use of the DC-DC converter at the turbine level. If one turbine within a string fails, the voltage is maintained by the other turbines in the string increasing their output voltage. This topology is designed to reduce the cable distance, but will undoubtedly require heavier and more expensive cables to utilise. Disadvantages of this topology are similar to those of the DC series topology.

## 6    Cost, Mass and Loss Estimation

The cost, mass and loss performance of each component used in the proposed topologies must be estimated in order to determine each topologies suitability. A variety of academic literature and commercial information has been used to accurately estimate the properties of each component used. Information regarding cost and mass of components is rarely available in the public domain for the exact power rating and size of components required. It is therefore necessary to rely on scaling relationships from the academic literature that estimate the cost and mass of generic components based on parameters like component power rating, $P$, or torque rating. Within this study, these scaling relationships have been adapted where possible to include th most up to date commercial information in order to reflect realistic components used within the wind industry. A summary of the relationships used to estimate mass and cost of components in this study is presented in Table 1. The cost of components can often be difficult to estimate as component producers will vary prices depending on market pressures, location





of projects, availability of materials etc. It is therefore necessary to rely on the academic literature in order to estimate the price of each component. Although the estimated prices may not be exact, they are sufficient to compare the cost effectiveness of each topology. All cost estimates used within this study are presented in 2019 GBP.

**Table 1.** Summary of mass and cost estimations of each component.

| Description | Mass (kg) | Cost (2019 GBP) |
|---|---|---|
| Medium speed PMSG | $7.78P^{0.9223}$ | $59.984P$ |
| 3 stage gearbox | $70.94 \times LSST^{0.759}$ | $18.033P^{1.249}$ |
| MV back-to-back VSC | $1.01P - 9.852$ | £$132/kVA$ |
| VSC (inv. or rec) | half of back-to-back VSC | half of back-to-back VSC |
| Bi-directional DC-DC converter | $(P/500) * 590$ | £$110/kVA$ |
| Cables (per km) | mass/km taken from data-sheets | $1000 \cdot (0.46 \times CSA + 94.671)$ |
| Transformers | $1.982P + 481.11$ | $-115,968 + 205.73P^{0.4473}$ |
| AC switchgear | N/A | $30,720 + 0.576V_{rated}$ |
| DC Switchgear | N/A | twice cost of AC switchgear |

## 6.1 Generators

NREL provide a mass estimation relationship for medium speed PMSG's in (Fingersh et al., 2006). However this relationship

was produced in 2006 and does not reflect any of the recent developments in generator design made over the last decade. Information is available from ABB about mass of generators in (ABB, 2012), which provides an up to date and industrial comparison point to check the NREL relationships against. Table 2 shows a comparison between actual mass given by ABB and the estimated mass using the NREL relationships from the 2006 study, as well as a correction factor required so that the two agree. It is clear that the original relationships overestimate the mass of PMSG's significantly, which reflects the advancement

in this technology in recent years. The most conservative correction factor has been applied to the relationship. This results in the relationship shown in Table 1, which also closely agrees with data given by The Switch in (The Switch, 2013). The cost of the PMSG's are also estimated using relationships provided by NREL in (Fingersh et al., 2006). Both mass and cost relationships are calculated based on the rated power of the machine in $kW$.

## 6.2 Gearboxes

All generators in this study are assumed to be used in connection with a three stage gearbox. This assumption is based on a typical rated speed of a medium speed PMSG of 1500 rpm and rated speed of the wind turbine rotors. This results in a required gearbox ratio of 1:50, which is easily achievable with a three stage gearbox. The cost and mass estimation relationship for a three stage gearbox is taken from the same NREL study as the generator relationships. It is assumed that the mass of gearboxes has not decreased considerably since the publication of this report as the gearbox was considered a mature technology in 2006.




**Table 2.** Comparison of NREL estimated mass and given mass of medium speed PMSG generators from ABB.

| Power rating (MW) | Given Weight (kg) | NREL estimate Weight | correction factor |
|---|---|---|---|
| 3 | 12,500 | 16,926 | 0.739 |
| 5 | 18,200 | 27,122 | 0.671 |
| 7 | 24,900 | 36,977 | 0.673 |

The mass of the gearbox is calculated based on the low speed shaft torque (LSST) and the cost is based on the rated power of the gearbox in $kW$

### 6.3 Power Electronic Converters

The mass of low voltage (690V AC), IGBT based back-to-back VSC's designed specifically for wind turbines are given for units of different power ratings in a data sheet provided by ABB (ABB, 2018). The mass is given for complete units and
includes the converters, filters, circuit breakers, casings, cooling systems and any other auxiliary systems that are required for the converters to operate. This information has been plotted and a linear approximation has been used to develop a scaling relationship that can estimate the mass of the back-to-back converter with power rating required for the MRWT system. Figure 10 shows that the linear approximation achieves a reasonable estimation of converter mass, particularly at lower power ratings. Some topologies also use medium voltage converters, but little information is available on the mass of commercially available
MV converters, particularly at low power ratings. Wind turbines have traditionally used LV generators of 690V until recently when power ratings of wind turbines have increased significantly and it became more appropriate to use MV machines. It can be assumed that the same relationship holds for both low and medium voltage back-to-back converters. Comparing the mass of LV and MV back-to-back converter units of similar power ratings, both produced by ABB, it is seen that the MV converter mass is lower compared to its LV counterpart (ABB, 2019), suggesting that the mass estimation for low power medium voltage
converters may be conservative. Mass for controlled rectifiers or inverters will be taken as half of the back-to-back converter mass.

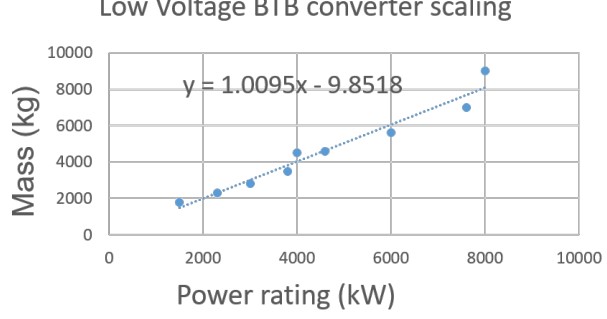

**Figure 10.** Scaling relationship for LV BTB converters.



Due to the variety of designs present in the literature and lack of commercially available high power DC-DC converters, estimates of mass are tricky. Mass estimates of various converter types are given in (Chen et al., 2013) which shows a mass of around 800kg for a buck/boost IGBT based 5 MW converter. The problem here is that this mass is solely based on the switches and passive elements in the DC-DC converter. When a commercially produced converter is available, the mass of the unit is likely to be considerably higher due to cooling and other auxiliary systems that are required. This lower estimate in mass could cause an unfair comparison between AC and DC topologies. Solar PV applications utilise DC-DC converters on the same scale as is required for the MRWT topologies. Dynapower provide some information about a commercially available bi-directional DC-DC converter in (Dynapower, 2019), which lists a mass of 500 $kW$ unit as 590 $kg$. Comparing this to the estimated mass of 800 $kg$ given in (Chen et al., 2013) for a 5 $MW$ unit highlights the difference between academic mass estimation and commercially available units. The Dynapower DC-DC converter can be used in a modular fashion to produce higher power DC-DC converters, so the mass of this unit will be used to estimate the mass of DC-DC converters required for this study. Mass of all power electronic devices in this study are estimated based on the rated power of the devices in $kW$.

Costs used in two recent studies (Parker and Anaya-Lara, 2013; Lakshmanan et al., 2015) have been converted to 2019 GBP to give a cost of a back-to-back converter (regardless of type) of £132/$kVA$ and a controlled rectifier of £66/$kVA$. DC-DC converter price estimates vary significantly in the literature due to the lack of commercially available converters for wind applications and the variety of designs suggested. It is therefore most appropriate to take a range of prices available in the literature for suitable DC-DC converters and take an average of this price. The studies used for this are: (Lakshmanan et al., 2015; Fingersh et al., 2006; Max et al., 2007; Lundberg, 2003; Georgios and Wheeler, 2010). This gives an average cost of £110/$kVA$.

### 6.4 Cables

Mass of cables can be easily assessed using mass per $km$ values given in data sheets of cables from a variety of cable manufacturers.

Cable costs are estimated from (Dicorato et al., 2011) which provides a cost function for a kilometre of cable based on the CSA of the cables. The study uses available costs from numerous sources and averages them for each CSA then uses a least square linear regression to produce the relationship. The paper states that this relationship is valid for medium voltage copper conductor XLPE insulated cables. Although not stated, most offshore installations use three core cable, so it will be assumed that this is the cost for three core cable. Suitable cables have been selected for each topology based on the required current carrying capacity of the cables. Cable distances have been estimated for each topology based on realistic cable layout designs that follow the triangular lattice support structure beams as closely as possible to avoid loose hanging cables within the structure.

### 6.5 Transformers

Mass of transformers has been estimated by using information given in (Declercq, 2003) regarding the 'SLIM' range of transformers for wind turbines manufactured by Pauwels International (now 'CG'). The mass of transformers from different





power ratings was plotted and a linear approximation for how the mass varies with power rating was developed. Additional information from (Islam et al., 2014) was used to develop the relationship shown in Table 1, which estimates the given masses of transformers very well. It also generally agrees with other information from data sheets for ABB distribution step down transformers. The mass of transformers is estimated based on the power rating of the transformer in $kVA$.

Lundberg (Lundberg, 2003) provides a formula for estimating the cost of MV/HV transformers rated between 6.3 and 150
MVA, with low side voltage rating between 10.5 and 77 kV and high side rating between 47kV and 140kV. The transformers required in this study do not quite fall into this category, but will be assumed to follow this cost model. This cost model provides the base for many cost estimates of transformers in the literature, with (Parker and Anaya-Lara, 2013; Dicorato et al., 2011; Lakshmanan et al., 2015) all using this relationship and scaling it to present day value in the currency of the paper. The cost estimates for transformers in this study are based on the rated power of the transformer in $VA$.

### 6.6 Switchgear

Lundberg (Lundberg, 2003) also provides a cost model for AC switchgear based on the voltage rating in $V$, which is commonly used within academic literature. The cost of DC switchgear is also required for this study, which is more difficult to estimate due to the lack of commercially available DC switchgear. It is suggested in (Lakshmanan et al., 2015) that the cost of DC switchgear could be as much as four times that of AC switchgear, but this figure is based on a study from 2007. Since then
advancements in DC switchgear have been made, so a relationship of two times the cost of AC switchgear will be used here to reflect advancements in technology. The mass of switchgear is assumed to be negligible in this study as the VSC mass estimations includes some switchgear masses.

### 6.7 Losses

Losses of each individual component were estimated at each 0.5 $m/s$ wind speed increment between cut-in and cut-out wind
speed. Mechanical power is calculated as follows

$$P_{mech} = \frac{1}{2}\rho A v^3 C_P \tag{1}$$

where $P_{mech}$ is the mechanical power produced by the turbine, $\rho$ is the density of air, $A$ is the swept area, $v$ is the wind speed and $C_P$ is the coefficient of performance. Loss profiles that describe the losses of each component over the entire operating range of wind speeds were used in order to account for varying efficiencies of components when operating at part load. For
each topology, the mechanical power was calculated using Eq (1) and losses for each component subtracted at each wind speed. The output power of each component was used as the input power of the next component. For example, the input power of the generator is the mechanical power produced by the turbine minus the gearbox losses. Matlab scripts were created to calculate the total losses at each wind speed of each collection topology.





An equation is presented for gearbox efficiency in (Jamieson, 2011), which was developed by GL Garrad Hassan. It comprises of a loss that varies in proportion to the operating power level, and a constant loss that is related to the rated power and number of stages:

$$L_{gear} = \frac{\left(\frac{10}{3} + 2N\right)P_r + 5NP_i}{1000} \tag{2}$$

where $L_{gear}$ is the losses in the gearbox in kW, $N$ is the number of stages, $P_r$ is the rated power of the gearbox in kW (constant) and $P_i$ is the input power of the gearbox in kW (variable). This equation is used to estimate losses of gearboxes in this study.

Losses for generators, power converters and DC-DC converters were estimated by using a combination of data presented in a Lundberg study (Lundberg, 2003) and various data sheets from commercial suppliers. The Lundberg study presents losses of various components as a percentage of rated power over a full range of wind speeds, so can be applied to a variety of different power ratings while the commercial data is used to ensure up to date losses at rated power of the different components are used. This allows for a simple method suitable for early stage analysis, while allowing the loss characteristics of different components at part load to be included. Loss data for each component was produced based on the Lundberg study and reconfigured to reflect commercial rated efficiencies and a different rated wind speed, while maintaining the below rated characteristics of the components.

Medium speed PMSG efficiency at rated power is listed as over 98% by ABB (ABB, 2012). Back-to-back converter efficiency at rated power for LV converter unit is given as 97% in (ABB, 2018). Back-to-back converter efficiency at rated power for MV converter unit is given as 98% in (ABB, 2019). These efficiencies include back-to-back converters, filtering, cooling systems etc. and is used in this study. For controlled rectifiers, an efficiency at rated power of 99% for MV and 98.5% for LV has been be used.

There are a huge amount of DC-DC converters proposed and studied in the literature in recent years. The advancement of power electronic devices and the increasing need to minimise mass in offshore applications has led to an increased interest in DC-DC converter topologies that are suitable for use in offshore wind farms. In (Parastar et al., 2015) a multilevel modular DC-DC converter for high voltage DC wind farm is suggested and reports efficiencies of around 97% at rated power for a small scale prototype. In (Chen et al., 2013) various types of converters are compared in terms of mass, number of components, efficiency and volume. The two best candidates are a buck/boost converter based on IGBT's with a rated efficiency of 97.5% and a resonant switched capacitor converter with a very high efficiency of over 99%. In (Max and Lundberg, 2008) three converter types are analysed and compared, with the author concluding that the most suited type of converter for use in a DC wind farm is a full bridge converter based on IGBT switches. The study presents efficiencies of both a turbine level and group level converter as 97.08% and 97.97% respectively. Finally, various topologies of DC turbine configurations are also shown in Lundberg's 2003 comprehensive study which all are approximately 97% efficiency (Lundberg, 2003). The conclusion reached is that a rated efficiency of 97% will be used for the DC-DC converters in this study, with below rated efficiencies capturing the characteristics of the converters presented in (Lundberg, 2003). This provides a realistic and slightly conservative estimate of losses within a DC-DC converter suitable for the MRWT application.



Losses in cables can be approximated by

$$L_{cable\_3\Phi} = 3I_{RMS}^2 R_{AC} \tag{3}$$

$$L_{cable\_DC} = 2I_{DC}^2 R_{DC} \tag{4}$$

where $L_{cable\_3\Phi}$ is the losses in the three phase AC cables in W, $I_{RMS}$ is the rms AC current in each phase in A, $R_{AC}$ is the resistance of the AC cable in $\Omega$, $L_{cable\_DC}$ is the losses in the DC cable in W, $I_{DC}$ is the DC current in A and $R_{DC}$ is the resistance of the DC cable in $\Omega$ (Starke et al., 2008). Resistances of the cables can be calculated by considering the total length of cables within the topology and the given resistance per length for the cables CSA.

     Transformer efficiencies are estimated using a combination of the loss profile presented by Lundberg in (Lundberg, 2003)

and full load efficiencies presented in (Islam et al., 2014). Losses in switchgear are considered negligible.

## 7   Cost effectiveness

In order to analyse the cost effectiveness of each topology a levelised cost of energy (LCOE) calculation was performed that considered the capital cost of the drive train and electrical components and the total electricity produced by each topology over a 20 year lifetime of the project in net present value (NPV). Equation 5 was used to calculate this LCOE and the NPV of

electricity produced was calculated according to Eq (6) where $AEC$ is the annual energy capture, $i$ is the discount rate which is equal to 10% and $n$ is the lifetime of the project in years. No costs associated with operations and maintenance or loss of power due to failures were included in this calculation.

$$LCOE_{elec} = \frac{\text{Capital cost of electrical components and gearbox}}{\text{sum of electricity produced over lifetime}} \tag{5}$$

$$NPV = \frac{AEC}{i}\left(1 - \frac{1}{(1+i)^n}\right) \tag{6}$$

The wind speed is assumed to follow a Weibull distribution and average wind speed of 10 $m/s$ is used. Total energy capture was calculated by multiplying the output power at each wind speed by the number of hours per year the wind will be at that speed and summing the results together.

## 8   Results and discussion

The total capital cost of each electrical topology is shown in Fig. 11. It is clear that DC topologies will incur a significantly

higher capital cost compared to AC topologies. This is due to the increased cost of DC-DC converters and DC switchgear.



The AC star and radial topologies have the lowest capital cost of topologies analysed, while the most expensive are the DC series/parallel and DC cluster topologies.

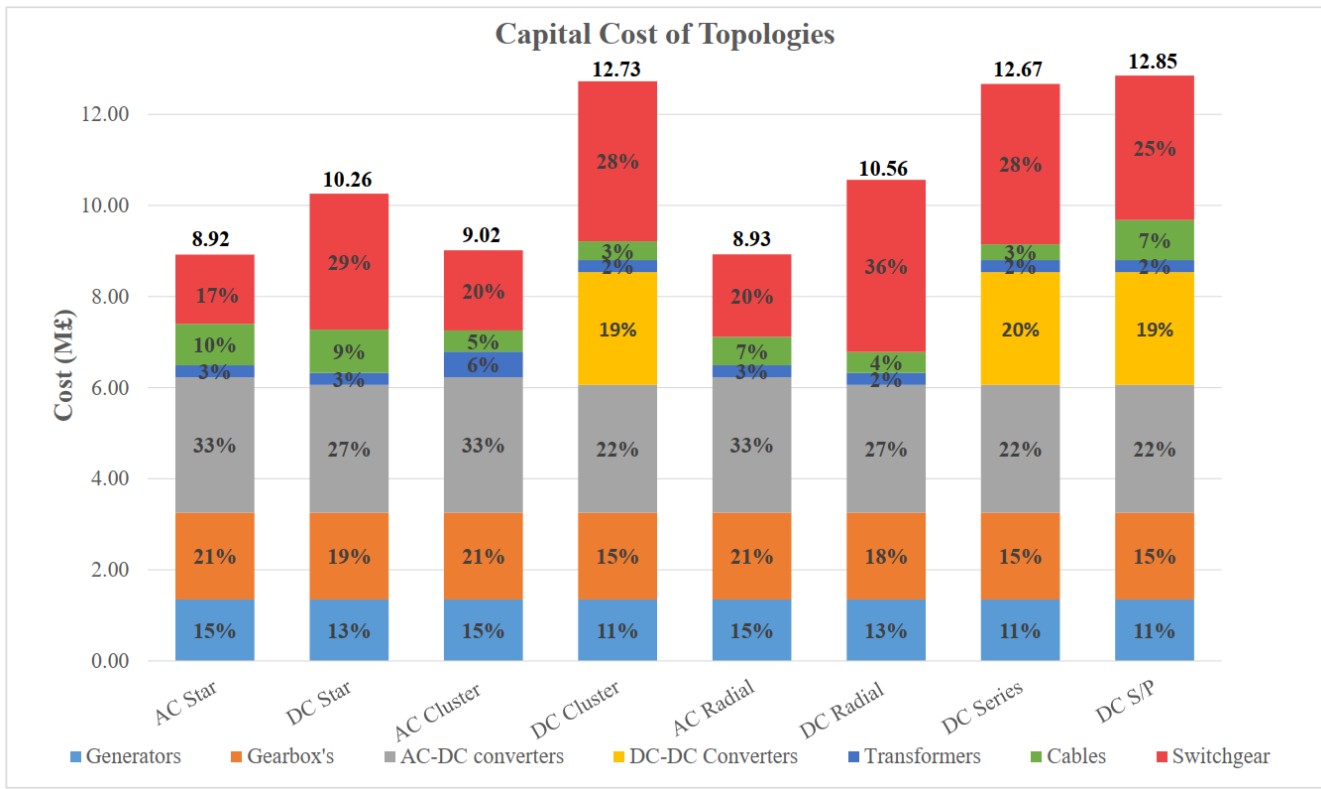

**Figure 11.** Total capital cost of electrical topologies with breakdown by component. Total capital cost is shown above each bar. Percentage of total capital cost is also shown for each component. Taking AC star as an example, the total capital cost is £8.92 million, with the generators accounting for 15% the total cost, gearboxes accounting for 21% and so on.

The total mass and breakdown of component mass is shown for each topology in Fig. 12. Although some topologies are lighter than others, the difference in total weight is not significant. The system with the highest mass is the AC cluster topology, with its high mass due to the use of AC transformers within the system. The lightest is the DC star topology, with DC radial and AC star very close behind. Gearbox and generator mass is a significant portion of total mass, accounting for 75-85% of the total mass in each case. This mass can be reduced by considering different generator and gearbox combinations, but has not been considered in this study so as to make a fair comparison of the type of electrical systems. As the large transformer and/or inverter are situated on the build-in platform of the MRWT, the mass per nacelle is a more appropriate measure of how the mass of each system will effect the design of the support structure. This is shown in Table 3 and includes all components housed within the nacelle as well as a share of the cable mass and shared components. The DC star and DC radial topologies

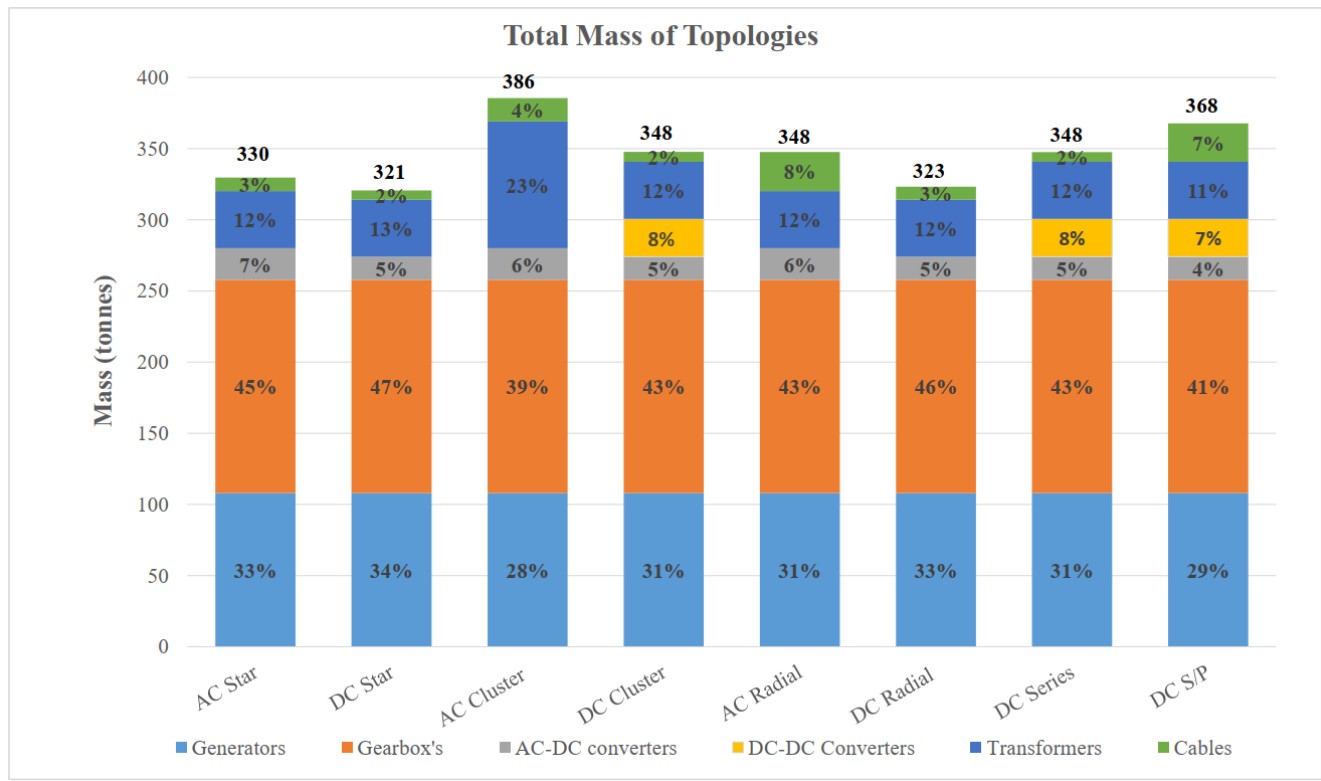

**Figure 12.** Total mass of electrical topologies with breakdown by component.

perform well in this category due to their low component count. The DC star achieves a 10.5% reduction in mass of each nacelle compared to the base case of the AC radial.

Losses at rated power of each topology are shown in Fig. 13, with the most efficient topologies being the DC radial and
AC radial designs. The DC radial design is slightly more efficient than the AC radial due to lower losses in the DC cables. This is due to the higher voltage level and the reduced conduction losses in DC cables. The DC cluster topology has higher losses compared to the AC cluster topology due to the much reduced efficiency of the DC-DC converters compared to AC transformers.

Comparing AC to DC equivalent systems the DC systems always have higher cost due to the high cost of DC-DC converters
and switchgear, with the mass of DC systems being marginally lower and the losses between AC and DC systems being very similar. This shows that the lower losses in DC cables at this power range and cable lengths are not large enough to make a significant difference in the overall efficiency of the systems. Comparing the AC and DC star topology efficiencies shows this well. Cables do not significantly contribute to the total mass, with the lightest system simply being the one with the fewest components.





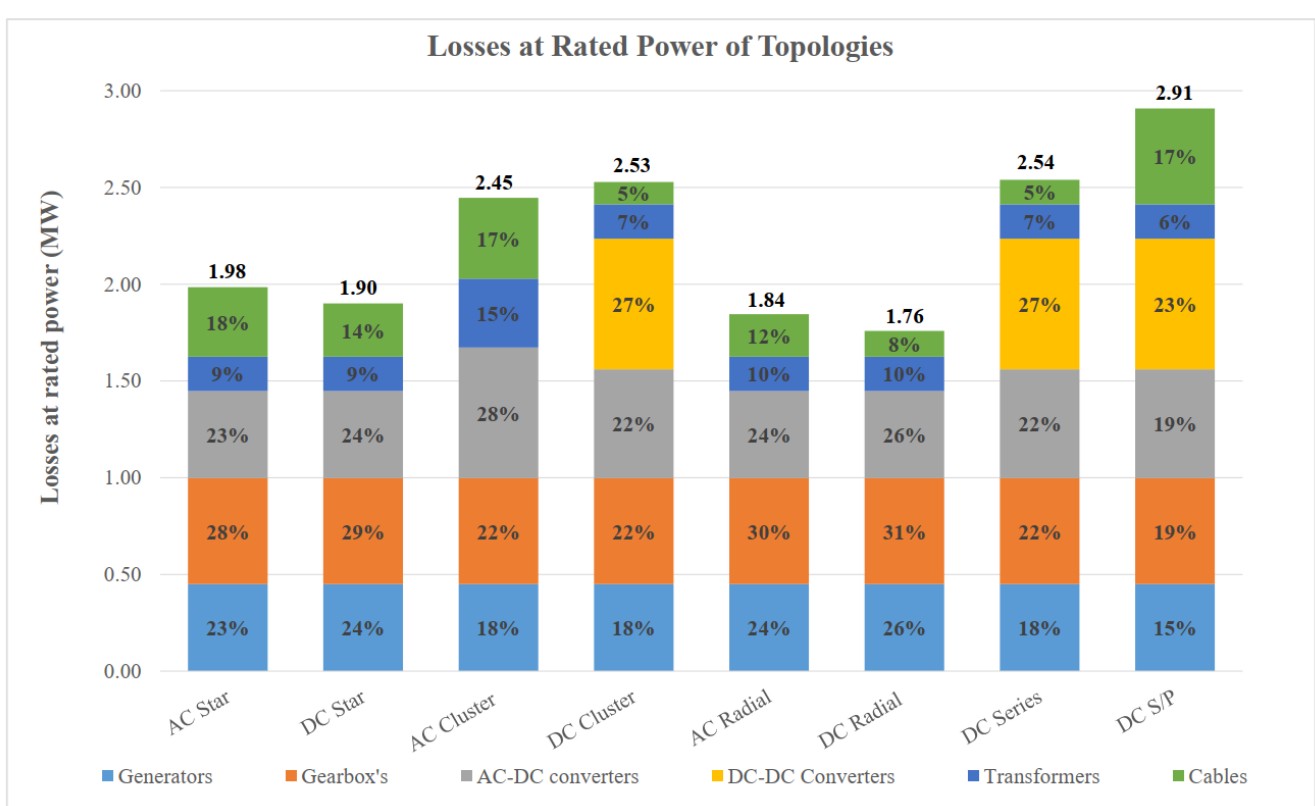

**Figure 13.** Total losses at rated power of electrical topologies with breakdown by component.

**Table 3.** Comparison of characteristics of each electrical topology.

| Topology (MW) | LCOE (£/MWh) | Mass per nacelle ($kg$) | Annual lost energy (GWh) |
|---|---|---|---|
| AC star | 12.62 | 6435 | 8.70 |
| DC star | 14.47 | 6117 | 8.45 |
| AC cluster | 13.06 | 7676 | 10.59 |
| DC cluster | 18.88 | 6721 | 10.94 |
| AC radial | 12.81 | 6834 | 8.28 |
| DC radial | 15.11 | 6177 | 8.02 |
| DC series | 18.80 | 6717 | 10.98 |
| DC series/parallel | 19.34 | 7167 | 12.05 |

The LCOE of the electrical system and gearbox of each topology is shown in Table 3. Low capital cost and losses in the AC systems results in low LCOE for the AC star, AC cluster and AC radial topologies. High capital cost and losses result in high LCOE for the DC series, DC series/parallel and DC cluster topologies. AC topologies have a lower capital cost and



lower LCOE compared to the DC systems analysed, showing that the proposed benefits of DC systems in offshore collection networks do not cross over to collection systems for MRWT's.

A comparison of each topology can be seen in Table 4. The AC radial is considered the base topology as it is the most commonly used collection network in offshore wind farms. Base levels are marked with 'O' in Table 4 with '+' symbols used to indicate an improvement in the specified category compared to the base case and '-' symbols used to indicate a decrease in performance in that category compared to the base case. Component count includes gearbox, generator, converters, transformers and switchgear units for each topology. The reliability category is considered as a combination of component

count and amount of shared equipment. As an example, the DC cluster topology performs badly in this category as a fault in the shared DC-DC converter would result in a loss of five turbines and the DC star topology performs excellently in this category due to having no shared equipment and a low component count. This category is seen as an important characteristic of each topology as it highlights one of the key benefits of a MRWT system. Failure of components in a large single rotor turbine will likely result in the loss of 100% of the power from that turbine, whereas a MRWT can still produce a high percentage of power

after the failure of components. Further work should be conducted in order to quantify monetary value of this characteristic for each topology, but a first pass attempt has been made here to at least indicate which topologies will benefit the most from this characteristic. Although cable failure rates are significantly lower than other components used in wind turbines, cable failures have still been included in this analysis. Cables in MRWT arrays will have less physical protection than sub sea cables used in wind farms so failure rates may be higher than stated in wind farm collection network failure studies.

The AC star topology is the best solution overall as it only performs slightly worse than the base case in the category of efficiency and performs very well in each of the other categories, specifically reliability and mass per nacelle. The DC star topology also performs well overall and could see an improvement in reliability compared to the AC star topology due to its lower component count. It is recommended that more detailed design and analysis work are carried out for these two topologies.

    The basic cost estimation of electrical components made in the Innwind project (Jamieson et al., 2015) totals to £4.6 million.

This cost estimation was lacking detail and significantly underestimated the cost of electrical systems required for a MRWT, with this study showing the cheapest system to cost in the region of £8.9 million. The capital cost of electrical components (as well as gearbox cost) in the Innwind project is 13.3% of the capital cost of the entire system, and LCOE of the electrical system and gearbox is estimated to be 10.65 $£/MWh$. Although lower than the LCOE of each topology in this study, it is reflective of the level of detail between the two studies. Using a similar methodology, the LCOE of the electrical system of two DTU

RWT's (inc. gearbox) is 36.85 $£/MWh$, showing that even the most expensive topologies are significantly cheaper compared to a large single rotor turbine. Combined with the savings in material costs for blades, improvements in O&M costs, reduced installation and transport costs and power increases due to clustering of turbines it is still expected that the MWRT concept will achieve a much improved overall LCOE regardless of the increase in cost of electrical components.

    The cost estimates presented here are highly sensitive to the cost of DC-DC converters and DC switchgear. Although these

costs may fall in the future, the reductions would have to be significant for the DC systems to be comparable in price to the AC systems in the context of MRWT electrical systems. DC systems may be more attractive if the MRWT would connect to a DC collection network, however there is currently no DC collection networks in operation, and the feasibility of their existence



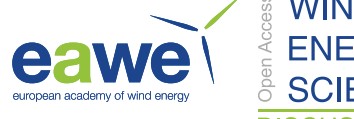

**Table 4.** Comparison of topologies.

| Topology | Cap. cost | Efficiency | LCOE | Total Mass | Mass per nacelle | Component count | Reliability | Notes |
|---|---|---|---|---|---|---|---|---|
| AC radial | O | O | O | O | O | O | O | O |
| DC radial | - | + | - | + | ++ | ++ | + | Shared cables (max. 6 turbines) but low component count |
| AC cluster | O | -- | O | - | -- | - | - | Shared transformer (5 turbines) and short shared cables |
| DC cluster | -- | -- | -- | O | + | + | -- | Shared DC-DC converter (5 turbines) and short shared cables |
| AC star | O | - | O | + | + | + | ++ | No shared equipment (high component count) |
| DC star | - | - | - | + | ++ | ++ | +++ | No shared equipment (low component count) |
| DC series | -- | -- | -- | O | + | + | -- | Shared DC-DC converter (5 turbines) and long shared cables |
| DC series/parallel | -- | -- | -- | - | - | O | -- | Long shared string cables, high current collection cables transmit all power, high component count |



within the near future is still low. DC systems could also reduce cost and mass significantly by using diode rectifier in place of controlled rectifiers, but this would jeopardise the controllability of the entire system and in most cases lose the ability to control the speed of the turbines individually.

## 9  Conclusions

Eight different electrical topologies have been proposed and analysed in terms of mass, cost effectiveness, component count and reliability. AC topologies consistently have the lowest LCOE of the electrical system ranging between 12.62 to 13.06 $\pounds/MWh$. DC systems have higher LCOE of the electrical system between 14.47 and 19.34 $\pounds/MWh$. The increased cost is attributed to the high cost of DC-DC converters and DC switchgear. The difference in total mass between the systems is not significant, but DC systems do see a reduction in mass per nacelle. It is unclear at this point how much the mass of each nacelle will effect the cost and complexity of the support structure required for a MWRT. DC-DC converters and DC switchgear are still commercially unavailable and it is therefore hard to estimate the cost of such equipment. Due to this, the results are very sensitive to the cost of DC-DC converters and switchgear. Systems that lose only a small portion of rated power due the failure of one component are clearly more desirable than systems that will lose a higher portion of rated power. Further work is recommended on this topic to investigate the monetary value of this characteristic of MRWT's.

From the systems analysed, the most promising type of system is the AC star topology; it performs well in a large range of categories, and has very good reliability compared to other systems. The high reliability is due to the design of the system which has no shared equipment between turbines. The DC star topology also performs well overall and could have even higher reliability than the AC star topology due to its lower component count. Both topologies should be considered for more detailed design and analysis in future work.

*Author contributions.*  PP carried out all design and analysis work under supervision and guidance from DC and OA.

*Competing interests.*  The authors declare that they have no conflict of interest

*Acknowledgements.*  This work was funded as part of the EPSRC-funded Wind and Marine Energy Systems Centre for Doctoral Training.



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
