# Peer review of "Comparison of electrical collection topologies for multi-rotor wind turbines"

_Wind Energy Science, 2020_

## Referee Comment (RC1) · Peter Dalhoff (Referee) · 27 Apr 2020

General comments:

The paper is well written. The topic of electrical topologies for MRWT's is novel and only little research has been published on this so far. The method is well described. Uncertainties and assumptions have been outlined clearly. The results lead to favourite design(s) and show a path where to do more research on. A sensitivity study would be beneficial to better understand the influence of the sometimes large uncertainties in component cost and mass.

Specific comments:

Line 24: I recommend "more and better material" instead of "more material" Line 94:

Why did you not use the same rated power as in Innwind (444 kW instead of 500 kW)? Is there a specific reason, e.g. power boost option? Line 108: How does the tower connect to the space frame? It would be good to understand from figure 1, whether the tower reaches to the top, to the bottom or somewhere in between the space frame. In line 133 and line 142 you mention the "base of the structure". What does it mean in particular and is it in conflict with line 108? Maybe it would be good to define the overall support structure and its elements (tower and space frame). A visualization of support structure and the electrical components within would be excellent.

Figure 8: Why are there only 4 wt's? Shouldn't it be 5?

Table 1: - For which power range is the table valid? - Some of the cost equations contain physical units, e.g. GBP/kVA and some contain variables, e.g. P. I recommend to keep it consistent - LSST in kNm? - Why do you call a 1500 rpm generator "medium speed". I thought this is high speed. Please clarify or cite a definition for medium and high speed Line 186: You name a 2012 paper "up to date". I understand that it is more up to date than the 2006 paper you refer on, but still it is not up to date in absolute terms.

Line 196: 1500 rpm is high speed in my opinion and needs 3 stage gearbox. Several hundred rpm is medium speed and needs 1 to 2 stages. Rotor speed is low speed and does not require gearbox. Maybe I am wrong with this definition, therefore please check and ideally cite a definition of high and medium speed.

Line 199: The torque density (kNm/kg) has been increased in recent years and is an ongoing research and optimization topic. The torque densities from 2006 to 2020 might have doubled approximately. As such your assumption "that the mass of gearboxes has not decreased considerably" might be wrong. Line 327: Wouldn't it be better then to call it "simplified cost of energy"? OPEX and losses are a huge factor. I guess any wind farm influences (cable losses, wake effects, HV station ) are not considered? I can understand that it is difficult to consider all these effects, but then it would be fair

to name them simplified CoE. Line 330: Whick k-factor of the Weibull distribution has been used?

Line 378: I would recommend not to make a guess here. E.g. in offshore wind farms, cables can and have been damaged by ships. This will not be the case for MRWT internal cables. Without having a database on root causes, I would recommend not to make guesses here.

Line 389: What do you mean by "similar"? Why did you not describe the approach to estimate the DTU-RWT cost? When you apply the same approach as you used for the MRWT, what will the result be for the DTU-RWT?

Table 4: Element AC star/Component count: here you give a "+" with high compenent count. At DC series/parallel you give a "0" with high component count? On which basis did you make the rankings and why the difference between the two?

Technical comments:

Line 2 (an others): MRWT Line 163: "it" Line 166: "string" Line 200 "shaft torque (LSST)" in kNm? Line 235 Why do you write kV A instead of kVA?

---

## Referee Comment (RC2) · Anonymous Referee #2 · 8 May 2020

This is generally an interesting topic. The research is entirely, especially now that MRWT is gaining attention. This study will provide the much needed insight into MRWT electrical system topologies and associated considerations. The manuscript is written and structured very well, apart from some occasional typos. However, there are some minor issues that need to be addressed before the paper is published.

1. Abstract: Although the abstract is well written, it would be good to provide some concluding remarks on findings from this research.

2. Introduction: P 2, Linen 41: It would be beneficial to provide further explanation on the relationship between "smoother load profile over MRWT structure" and the fatigue of MRWT. Line 41 – 43: In addition, the claim that "Various studies performed at Kyushu University in Japan (Göltenbott et al., 2017; Ohya et al., 2017; Goeltenbott et

al., 2015) have shown that clustering turbines together can improve their performance"
needs to be more specific in the context of this paper by stating the actual performance
components that has been enhanced. P3, Line 64: Typo – Section 2 "describes" ...

3. Design constraints: It would be most appropriate if the properties of the models used
in this study (10MW and MRWT) are summarised in a table for clarity. Lines 98 – 100:
there appears to be something missing in "diameter or 41 m. . ..". Please crosscheck.
Again, these values are best presented in a table and they should be supported with
reference(s). Notwithstanding the assumptions made, could provide some details on
the platform? This is important even if does not support the transformers/ converters.

4. Proposed electrical topology designs: The authors have based some assumptions
on the topology design on common offshore collection network. Does this mean that
this method is applicable to offshore MRWT concept? If yes, how does it handle the
effect of offshore environment on each topology? Also, going by the assumption that
the "base of the structure", which I assume is the foundation; since offshore environ-
ment may not have space to accommodate the transformer/converter at the "base of
the structure", what impact will this have on these topologies?

A graphic illustration of the platform, rather the textual description, showing the electri-
cal component is recommended. It would good if you could provide recommendations
on how to mitigate the noted disadvantages of each cluster topology or otherwise.

5. Cost, Mass and Loss Estimation: Typo in Line 178, . . .include "the" . . . Costs esti-
mate has been presented in Table 1 without stating the type of wind turbine for which
it applies. Please clarify. There are some inconsistencies in the use of units. Please
check carefully.

Gearbox rpm: You may need to update your speed regime classification. Typically, 1.
Slow-Speed: rated speed less than 400 rpm. 2. Medium Speed: rated speed of 400
rpm to 1200 rpm. 3. High-Speed: rated speed of 1200 rpm or more.

[Figure]

Power electronic converter: Although the estimation of converter mass shows a linear scaling relationship for LV - BTB converters, but this relationship is only limited to low power ratings. Any thought how this would affect the topologies performance and its commercial viability?

Cables (Line 236): Cable damage is a common occurrence in offshore environment; there is need to consider this in the topology design if this study is to be applied of offshore MRWT.

Transformers (Line 248): There is high inconsistent use of units in this section. Please crosscheck all units carefully.

6. The conclusions should be improved to include summary of key findings and their impact on MRWT. Although a good investigation of the various components of electrical topologies design have been undertaken, the results have not been completely validated in terms of their optimal values and their impact on the overall performance on MRWT. It is suggested that the authors should consider this doing it

---

## Author Comment (AC1) · 12 Jun 2020

In this comment the authors of the paper will take the opportunity to address the comments made my the referees. The comments by the referees will be stated in quotation marks and then followed by the response from the authors.

The authors would like to thank both referees for taking the time to review this paper and make suggestions for improvement.

First review – Peter Dalhoff

General Comments:

"The paper is well written. The topic of electrical topologies for MRWT's is novel and

only little research has been published on this so far. The method is well described. Uncertainties and assumptions have been outlined clearly. The results lead to favourite design(s) and show a path where to do more research on. A sensitivity study would be beneficial to better understand the influence of the sometimes large uncertainties in component cost and mass."

The authors are grateful with the reviewer for the time and effort spent in reviewing our work. We now proceed to answer the reviewer comments individually.

Specific Comments:

1. "Line 24: I recommend "more and better material" instead of "more material""

Agreed, this has been added to the revised version of the paper

2. "Line 94: Why did you not use the same rated power as in Innwind (444 kW instead of 500 kW)? Is there a specific reason, e.g. power boost option?"

Although the Innwind project uses 444kW turbines, the authors believe that the same swept area could realistically produce 500 kW given that rotors of this size produce 500 kW in industry standard turbines like the Enercon E40 turbine (500 kW, rotor diameter of 40.3m) and the Vestas V39 turbine (500 kW, rotor diameter of 39m). The authors believe that this gives a fairer representation of the energy capture possible with such a system. An additional sentence has been added to the revised version of the paper to clarify this to the reader.

3. "Line 108: How does the tower connect to the space frame? It would be good to understand from figure 1, whether the tower reaches to the top, to the bottom or somewhere in between the space frame. In line 133 and line 142 you mention the "base of the structure". What does it mean in particular and is it in conflict with line 108? Maybe it would be good to define the overall support structure and its elements (tower and space frame). A visualization of support structure and the electrical components within would be excellent."

Thanks for your suggestion. In order to provide better understanding of the support structure of the system, a diagram that describes the overall structure of the system has been included in the revised version of the paper (figure 2) and an additional paragraph has been added to section 3 to better describe the system.

4. "Figure 8: Why are there only 4 wt's? Shouldn't it be 5?"

Thanks for the observation, Figure 8 has been amended to include 5 turbines in the revised version of the paper.

5. "Table 1: - For which power range is the table valid? - Some of the cost equations contain physical units, e.g. GBP/kVA and some contain variables, e.g. P. I recommend to keep it consistent - LSST in kNm? - Why do you call a 1500 rpm generator "medium speed". I thought this is high speed. Please clarify or cite a definition for medium and high speed Line 186: You name a 2012 paper "up to date". I understand that it is more up to date than the 2006 paper you refer on, but still it is not up to date in absolute terms."

Valid range of power ratings vary from component to component. To make this clearer for the reader an additional column has been included in table 1 to indicate the range in which the relationships are valid for each component.

Equations are presented in a consistent manner throughout the table in the revised version of the paper – equations containing units have been changed to contain variables.

Yes, LSST in kNm, this has been included in the revised version of the paper.

You are correct that 1500 rpm is considered high speed. The original cost relationship used is for medium speed generators – this relationship has been adapted to a relationship for high speed generators. This does not affect the final results as each system uses the same type of generator.

With regards to referring to a 2012 paper as up to date – this is a fair comment. The

revised version of the paper makes it clear that the information presented is taken from ABB website and is presented there as their current technology. Thanks for these observations.

6. "Line 196: 1500 rpm is high speed in my opinion and needs 3 stage gearbox. Several hundred rpm is medium speed and needs 1 to 2 stages. Rotor speed is low speed and does not require gearbox. Maybe I am wrong with this definition, therefore please check and ideally cite a definition of high and medium speed."

You are correct with this. This has been amended in the revised version of the paper.

7. "Line 199: The torque density (kNm/kg) has been increased in recent years and is an ongoing research and optimization topic. The torque densities from 2006 to 2020 might have doubled approximately. As such your assumption "that the mass of gearboxes has not decreased considerably" might be wrong. "

Thanks for your observations. The authors agree that this statement requires more clarification and research. However, a different value of generator mass will not affect the results presented in the paper as each topology uses the same gearbox. The issue of sizing of gearboxes is an interesting topic that will be addressed in future research.

8. "Line 327: Wouldn't it be better then to call it "simplified cost of energy"? OPEX and losses are a huge factor. I guess any wind farm influences (cable losses, wake effects, HV station ) are not considered? I can understand that it is difficult to consider all these effects, but then it would be fair to name them simplified CoE. "

Yes, this a fair comment and the revised version of the paper has been updated accordingly. Thank you for this suggestion as it is a good way of clarifying the difference between true LCOE and the cost effectiveness figures used in this paper for comparison.

9. "Line 330: Whick k-factor of the Weibull distribution has been used?"

Values of k=2 and C=9.5 have been used. The revised version of the paper has included these values in section 7.

10. "Line 378: I would recommend not to make a guess here. E.g. in offshore wind farms, cables can and have been damaged by ships. This will not be the case for MRWT internal cables. Without having a database on root causes, I would recommend not to make guesses here."

The authors agree with your suggestion. This statement has been omitted from the revised version of the paper.

11. "Line 389: What do you mean by "similar"? Why did you not describe the approach to estimate the DTU-RWT cost? When you apply the same approach as you used for the MRWT, what will the result be for the DTU-RWT?"

The authors agree that using the term "similar" is not ideal here. After checking over the analysis, the methods were in fact the same. This has been corrected in the revised version of the paper.

12. "Table 4: Element AC star/Component count: here you give a "+" with high component count. At DC series/parallel you give a "0" with high component count? On which basis did you make the rankings and why the difference between the two?"

The base case of AC radial has a high component count compared to the rest of the topologies. The DC series/parallel has the same component count as the AC radial. The only topology with a higher component count than the AC radial and the DC series/parallel is the AC cluster. The high component count comment for the AC star was to distinguish the difference in reliability between AC and DC star, and to highlight that the DC star will have better reliability due to using less components. In order to clarify the matter, the AC star note has been changed to "medium component count" in the revised version of the paper, and a note has been added to the base case AC radial to indicate that it has a high component count also. Thank you for the observation.

"Technical comments: Line 2 (an others): MRWT Line 163: "it" Line 166: "string" Line

200 "shaft torque (LSST)" in kNm? Line 235 Why do you write kV A instead of kVA?"

All technical comments and typos have been corrected in the revised version of the paper.
* * *
* * *
Anonymous Referee #2

General Comments:

"This is generally an interesting topic. The research is entirely, especially now that MRWT is gaining attention. This study will provide the much needed insight into MRWT electrical system topologies and associated considerations. The manuscript is written and structured very well, apart from some occasional typos. However, there are some minor issues that need to be addressed before the paper is published"

The authors are grateful with the reviewer for the time and effort spent in reviewing our work. We now proceed to answer the reviewer comments individually.

1. "Abstract: Although the abstract is well written, it would be good to provide some concluding remarks on findings from this research."

Following the reviewers advice, the authors have included some of the findings of our research in the abstract section

2. "Introduction: P 2, Linen 41: It would be beneficial to provide further explanation on the relationship between "smoother load profile over MRWT structure" and the fatigue of MRWT. "

Thanks for your suggestion. Larger load variations lead to more fatigue damage, and this must be accounted for when designing the support structure. Loading is generally highest at rated wind speed (just before blades start to pitch). It is highly unlikely that all individual rotors will be operating at rated wind speed at the same time due

to the nature of wind. This results in a load averaging effect over the MRWT support structure, meaning that loading over time is smoother and has less variation. Support structure elements will be subjected smaller variations in loading over the life time of the structure, which will lead to less degradation in the structural components. An additional paragraph has been added to the section clarify this this.

3. "Line 41 – 43: In addition, the claim that "Various studies performed at Kyushu University in Japan (Göltenbott et al., 2017; Ohya et al., 2017; Goeltenbott et al., 2015) have shown that clustering turbines together can improve their performance" needs to be more specific in the context of this paper by stating the actual performance components that has been enhanced. "

Thanks for the suggestion. The studies show that there is an increase in Cp when clustering turbines together, which will lead to an increase in annual energy capture. An additional paragraph has been added to the revised version of the paper to clarify this

4. "P3, Line 64: Typo – Section 2 "describes" ..."

Corrected.

5. "Design constraints: It would be most appropriate if the properties of the models used in this study (10MW and MRWT) are summarised in a table for clarity. "

Following the reviewers advice, a table will be included in section 3 of the revised version of the paper to summarise these properties.

6. "Lines 98 – 100: there appears to be something missing in "diameter or 41 m. . ..". Please crosscheck. Again, these values are best presented in a table and they should be supported with reference(s)."

Thanks for the observation. This is a typo and should read "diameter of 41 m". ". A new table summarising these properties has been included to the revised version of the paper.

7. "Notwithstanding the assumptions made, could provide some details on the platform? This is important even if does not support the transformers/ converters."

An assumption made about the MRWT concept in previous studies is that there will be a small platform attached to the structure below the array of wind turbines but above sea level that is big enough to support any electrical equipment needed, for instance a large step up transformer. In order to provide more details on the platform, a new diagram has been included in the revised version of the paper (figure 2).

8. "Proposed electrical topology designs: The authors have based some assumptions on the topology design on common offshore collection network. Does this mean that this method is applicable to offshore MRWT concept? If yes, how does it handle the effect of offshore environment on each topology? Also, going by the assumption that the "base of the structure", which I assume is the foundation; since offshore environment may not have space to accommodate the transformer/converter at the "base of the structure", what impact will this have on these topologies? A graphic illustration of the platform, rather the textual description, showing the electrical component is recommended."

The authors have drawn from knowledge in the literature regarding offshore collection network design, as it is analogous to the electrical system required for MRWT's. Both have the purpose of collecting power from a number of turbines and exporting it to a wider electrical network at one single point of connection. The designs proposed in the paper are applicable to both onshore and offshore MRWT's. The effects of an offshore environment on components is an interesting topic, however, this is out of the scope of this project as it will not affect the results of the study. Large wind turbines are satisfactorily protected from the offshore environments so it is assumed that MRWT nacelles are also satisfactorily protected.

The base of the structure is misleading and an illustration has been included to clarify this in the revised version of the paper (figure 2). By base of the structure, the authors

refer to a platform situated below the array of wind turbines but above sea level which will be part of the overall structure of the MRWT.

9. "It would good if you could provide recommendations on how to mitigate the noted disadvantages of each cluster topology or otherwise."

Thanks for the suggestion; unfortunately, the disadvantages of the cluster topologies are inherent in the design of those topologies. There is no real way to mitigate them without changing the design so they are no longer cluster topologies. Because of this reason, we provide the reader with several topologies to help evaluate the pros and cons of each system.

10. "Cost, Mass and Loss Estimation: Typo in Line 178, . . .include "the" . . . "

Corrected

11. "Costs estimate has been presented in Table 1 without stating the type of wind turbine for which it applies. Please clarify."

Thanks for the observation. In order to clarify, a sentence has been added in the revised version of the paper to inform the reader that these cost estimates have been made for pitch regulated, variable speed wind turbines.

12. "There are some inconsistencies in the use of units. Please check carefully."

Table 1 will be updated so all cost and mass functions use variables instead of units in the formula. The units have been amended and are expressed consistently where possible in the revised version of the paper.

13. "Gearbox rpm: You may need to update your speed regime classification. Typically, 1. Slow-Speed: rated speed less than 400 rpm. 2. Medium Speed: rated speed of 400 rpm to 1200 rpm. 3. High-Speed: rated speed of 1200 rpm or more. "

The reviewer is correct in mentioning that a 1500 rpm is considered high speed. The original cost relationship used is for medium speed generators – this relationship has

been adapted to a relationship for high speed generators in the revised version of the paper. This does not affect the final results as each system uses the same type of generator.

14. "Power electronic converter: Although the estimation of converter mass shows a linear scaling relationship for LV - BTB converters, but this relationship is only limited to low power ratings. Any thought how this would affect the topologies performance and its commercial viability?"

With regards to the converter mass scaling, Figure 10 shows that the linear relationship holds well for power ratings up to 6 MW. Higher power ratings still show a linear relationship, but there is slightly higher variance above 6 MW. Higher ratings are unlikely to be required in MRWTs. In view of this, the authors do not believe this would affect the commercial viability of the MRWT.

15. "Cables (Line 236): Cable damage is a common occurrence in offshore environment; there is need to consider this in the topology design if this study is to be applied of offshore MRWT. "

Although cable damage is common in offshore wind farms, cable failure rates are typically much lower compared to other components in an offshore wind farm, like generator and converter failures. The topic of reliability analysis for MRWT's is a very interesting and important topic, but detailed work on this subject is out of the scope of this study. The paper does include a qualitative analysis on reliability, which is highlighted in the comparison table presented at the end (Table 4). Cable failures are considered in this at least in a qualitative manner and in terms of how many turbines would be lost because of a cable failure. A recommendation for future work on the topic of MRWT reliability is recommended in the revised version of the paper as the authors recognize it is an important area for investigation.

16. "Transformers (Line 248): There is high inconsistent use of units in this section. Please crosscheck all units carefully."
[Figure]

Mass estimates are based on power rating in kVA, and cost estimates used are based on VA. This has been changed in the revised version of the paper so both units are consistent. Thanks for the observation.

17. "The conclusions should be improved to include summary of key findings and their impact on MRWT. Although a good investigation of the various components of electrical topologies design have been undertaken, the results have not been completely validated in terms of their optimal values and their impact on the overall performance on MRWT. It is suggested that the authors should consider this doing it"

Thank you for these comments. A summary of key findings has been included in the conclusion of the revised version of the paper. The authors recognize that numerous parameters like voltage level, number of turbines in clusters and different electrical configurations within the topologies should be considered in future work, but is out of the scope of this paper. An additional paragraph has been added to the end of section 8 in which the authors make recommendations for future work on these topics. The authors believe that this paper provides insight into the effectiveness of various topology designs, and although not all parameters are fully optimised, the work can form a base for future projects and full optimisation of the system.

---

## Author Response (AR2)

**Comparison of electrical collection topologies for multi-rotor wind turbines**

Paul Pirrie[1], David Campos-Gaona[1], and Olimpo Anaya-Lara[1]

[1]Wind and Marine Energy Systems CDT, University of Strathclyde, Royal College Building, 204 George Street, Glasgow, Scotland

**Correspondence:** Paul Pirrie (paul.pirrie@strath.ac.uk)

**Authors response**

The following section outlines the authors response to reviewers questions. The following layout will be used throughout:

1. Reviewers comment

    – Authors response

    – Action taken by author in revised manuscript

**All line numbers refer to the marked up version of the manuscript. These may vary slightly from the revised manuscript as the marked up version includes deleted text.**

**Authors response to first reviewer - Peter Dalhoff**

1. The paper is well written. The topic of electrical topologies for MRWT's is novel and only little research has been published on this so far. The method is well described. Uncertainties and assumptions have been outlined clearly. The results lead to favourite design(s) and show a path where to do more research on. A sensitivity study would be beneficial to better understand the influence of the sometimes large uncertainties in component cost and mass.

    – The authors are grateful with the reviewer for the time and effort spent in reviewing our work. We now proceed to answer the reviewer comments individually.

    – No action required

2. Line 24: I recommend 'more and better material' instead of 'more material'

    – Agreed, this has been added to the revised version of the paper

    – Added 'and better' to line 31

3. Line 94: Why did you not use the same rated power as in Innwind (444 kW instead of 500 kW)? Is there a specific reason, e.g. power boost option?

– Although the Innwind project uses 444kW turbines, the authors believe that the same swept area could realistically produce 500 kW given that rotors of this size produce 500 kW in industry standard turbines like the Enercon E40 turbine (500 kW, rotor diameter of 40.3m) and the Vestas V39 turbine (500 kW, rotor diameter of 39m). The authors believe that this gives a fairer representation of the energy capture possible with such a system. An additional sentence has been added to the revised version of the paper to clarify this to the reader.

– Added 3 sentences for clarification of this point. Starting line 118.

4. Line 108: How does the tower connect to the space frame? It would be good to understand from figure 1, whether the tower reaches to the top, to the bottom or somewhere in between the space frame. In line 133 and line 142 you mention the "base of the structure". What does it mean in particular and is it in conflict with line 108? Maybe it would be good to define the overall support structure and its elements (tower and space frame). A visualization of support structure and the electrical components within would be excellent.

– Thanks for your suggestion. In order to provide better understanding of the support structure of the system, a diagram that describes the overall structure of the system has been included in the revised version of the paper (Figure 1) and an additional paragraph has been added to section 3 to better describe the system.

– Figure 1 has been adapted to show the physical layout of the system, and a further description of the system has been added at the start of section three to clarify the support structure design. Line 103 - 112.

5. Figure 8: Why are there only 4 wt's? Shouldn't it be 5?

– Thanks for the observation, Figure 8 has been amended to include 5 turbines in the revised version of the paper.

– Figure 8 amended.

6. Table 1: - For which power range is the table valid?

– Valid range of power ratings vary from component to component. To make this clearer for the reader an additional column has been included in Table 1 (now Table 2) to indicate the range in which the relationships are valid for each component.

– Additional column added to Table 2 (Summary of mass and cost estimations of each component) in which the valid range of each component is stated. (marked up version of manuscript has issue displaying this table properly. Please see proper table in revised manuscript pdf.)

7. Table 1 - Some of the cost equations contain physical units, e.g. GBP/kVA and some contain variables, e.g. P. I recommend to keep it consistent - LSST in kNm?

– Equations are presented in a consistent manner throughout the table in the revised version of the paper – equations containing units have been changed to contain variables.
Yes, LSST in kNm, this has been included in the revised version of the paper.

– Table 2 (previously Table 1) has been amended so all equations are based on variables and do not include units. Equations have also been simplified where possible but are still the same relationships. LSST has been stated to be in kNm in line 252.

8. Why do you call a 1500 rpm generator "medium speed". I thought this is high speed. Please clarify or cite a definition for medium and high speed Line

   – You are correct that a 1500 rpm is considered high speed. The original cost relationship used is for medium speed generators – this relationship has been adapted to a relationship for high speed generators. This does not affect the final results as each system uses the same type of generator.

   – The relationship used for cost and mass estimation has been adapted to be accurate for high speed generators instead of medium speed generators. Section 6.1 (Generators) has been rewritten to describe this method. Information in Table 2 has also been updated to reflect the changes made. All mentions of medium speed generators have been changed to high speed generator.

9. Line 186: You name a 2012 paper "up to date". I understand that it is more up to date than the 2006 paper you refer on, but still it is not up to date in absolute terms.

   – With regards to referring to a 2012 paper as up to date – this is a fair comment. The revised version of the paper makes it clear that the information presented is taken from ABB website and is presented there as their current technology. Thanks for these observations.

   – This section is no longer in the manuscript due to the change in section 6.1. However, it is also mentioned later in section 6.7 (Losses) - an additional sentence has been added to clarify that this is ABB current technology - line 350.

10. Line 196: 1500 rpm is high speed in my opinion and needs 3 stage gearbox. Several hundred rpm is medium speed and needs 1 to 2 stages. Rotor speed is low speed and does not require gearbox. Maybe I am wrong with this definition, therefore please check and ideally cite a definition of high and medium speed.

    – You are correct with this. This has been amended in the revised version of the paper.

    – This has been amended in the revised manuscript - see item 8.

11. Line 199: The torque density (kNm/kg) has been increased in recent years and is an ongoing research and optimization topic. The torque densities from 2006 to 2020 might have doubled approximately. As such your assumption "that the mass of gearboxes has not decreased considerably" might be wrong.

    – Thanks for your observations. The authors agree that this statement requires more clarification and research. However, a different value of generator mass will not affect the results presented in the paper as each topology uses the same gearbox. The issue of sizing of gearboxes is an interesting topic that will be addressed in future research.

– Sentence added for clarification starting line 250.

12. Line 327: Wouldn't it be better then to call it "simplified cost of energy"? OPEX and losses are a huge factor. I guess any wind farm influences (cable losses, wake effects, HV station ) are not considered? I can understand that it is difficult to consider all these effects, but then it would be fair to name them simplified CoE

    – Yes, this a fair comment and the revised version of the paper has been updated accordingly. Thank you for this suggestion as it is a good way of clarifying the difference between true LCOE and the cost effectiveness figures used here for comparison.

    – All instances of LCOE in the manuscript have been changed to simplified COE.

13. Line 330: Which k-factor of the Weibull distribution has been used?

    – Values of k=2 and C=9.5 have been used. The revised version of the paper has included this values in section 7

    – Values added to line 387.

14. Line 378: I would recommend not to make a guess here. E.g. in offshore wind farms, cables can and have been damaged by ships. This will not be the case for MRWT internal cables. Without having a database on root causes, I would recommend not to make guesses here.

    – The authors agree with your suggestion. This statement has been omitted from the revised version of the paper.

    – Sentence omitted from line 444.

15. Line 389: What do you mean by "similar"? Why did you not describe the approach to estimate the DTU-RWT cost? When you apply the same approach as you used for the MRWT, what will the result be for the DTU-RWT?

    – The authors agree that using the term "similar" is not ideal here. After checking over the analysis, the methods were in fact the same. This has been corrected in the revised version of the paper.

    – 'Similar' replaced with 'the same' in line 455.

16. Table 4: Element AC star/Component count: here you give a "+" with high component count. At DC series/parallel you give a "0" with high component count? On which basis did you make the rankings and why the difference between the two?

    – The base case of AC radial has a high component count compared to the rest of the topologies. The DC series/parallel has the same component count as the AC radial. The only topology with a higher component count than the AC radial and the DC series/parallel is the AC cluster. The high component count comment for the AC star was to distinguish the difference in reliability between AC and DC star. In order to clarify the matter, the AC star note has been changed to "medium component count" in the revised version of the paper, and a note has been added to the base case AC radial to indicate that it has a high component count also. Thank you for the observation.

– Table 4 - Notes column amended to include 'high component count' for AC radial, 'medium component count' for AC star and 'high component count' for AC cluster.

17. Line 2 (an others): MRWT

    Line 163: "it"

    Line 166: "string"

    Line 200 "shaft torque (LSST)" in kNm?

    Line 235 Why do you write kV A instead of kVA?

    – All typos have been corrected in the revised version of the paper

    – corrected in:

        – line 2

        – line 201

        – line 252

        – line 292

**Authors response to second reviewer - Anonymous Referee 2**

1. This is generally an interesting topic. The research is entirely, especially now that MRWT is gaining attention. This study will provide the much needed insight into MRWT electrical system topologies and associated considerations. The manuscript is written and structured very well, apart from some occasional typos. However, there are some minor issues that need to be addressed before the paper is published

    – The authors are grateful with the reviewer for the time and effort spent in reviewing our work. We now proceed to answer the reviewer comments individually.

    – No action required

2. Abstract: Although the abstract is well written, it would be good to provide some concluding remarks on findings from this research.

    – Following the reviewers advice, the authors have included some of the findings of our research in the abstract section

    – Abstract extended to include more key findings. Line 13 to line 21.

3. Introduction: P 2, Line 41: It would be beneficial to provide further explanation on the relationship between "smoother load profile over MRWT structure" and the fatigue of MRWT.

– Thanks for your suggestion. Larger load variations lead to more fatigue damage, and this must be accounted for when designing the support structure. Loading is generally highest at rated wind speed (just before blades start to pitch). It is highly unlikely that all individual rotors will be operating at rated wind speed at the same time due to the nature of wind. This results in a load averaging effect over the MRWT support structure, meaning that loading over time is smoother and has less variation. Support structure elements will be subjected to smaller variations in loading over the life time of the structure, which will lead to less degradation in the structural components. An additional paragraph has been added to the section to clarify this.

– Section of text added to clarify this point. Line 49 to line 55.

4. Line 41 – 43: In addition, the claim that "Various studies performed at Kyushu University in Japan (Göltenbott et al., 2017; Ohya et al., 2017; Goeltenbott et al., 2015) have shown that clustering turbines together can improve their performance" needs to be more specific in the context of this paper by stating the actual performance components that has been enhanced.

– The studies show that there is an increase in Cp when clustering turbines together, which will lead to an increase in annual energy capture. An additional sentence has been added to the revised version of the paper to clarify this

– Changed sentence to read "Various studies performed at Kyushu University in Japan (Göltenbott et al., 2017; Ohya et al., 2017; Goeltenbott et al., 2015) have shown that clustering turbines together can result in an increase the coefficient of power, $C_P$, which can lead to increases in annual energy capture. Line 57.

5. P3, Line 64: Typo – Section 2 "describes" ...

– Corrected.

– Corrected - line 79

6. Design constraints: It would be most appropriate if the properties of the models used in this study (10MW and MRWT) are summarised in a table for clarity.

– Following the reviewers advice, a table will be included in section 3 of the revised version of the paper to summarise these properties.

– Table 1 added to revised version of manuscript.

7. Lines 98 – 100: there appears to be something missing in "diameter or 41 m. . ..". Please crosscheck. Again, these values are best presented in a table and they should be supported with reference(s).

– Thanks for the observation. This is a typo and should read "diameter of 41 m". ". A new table summarising these properties has been included to the revised version of the paper.

– Sentence omitted due to this detail being included in Table 1 in the revised version of the manuscript. Line 130

8. Notwithstanding the assumptions made, could provide some details on the platform? This is important even if does not support the transformers/ converters.

   – An assumption made about the MRWT concept in previous studies is that there will be a small platform attached to the structure between the top of the tower and the base of the wind turbine array. In order to provide more details on the platform, a new diagram (Figure 1) has been included in the revised version of the paper.

   – Figure 1 has been adapted to show the physical layout of the system, and a further description of the system has been added at the start of section three to clarify the support structure design. Line 103 to line 112.

9. Proposed electrical topology designs: The authors have based some assumptions on the topology design on common offshore collection network. Does this mean that this method is applicable to offshore MRWT concept? If yes, how does it handle the effect of offshore environment on each topology? Also, going by the assumption that the "base of the structure", which I assume is the foundation; since offshore environment may not have space to accommodate the transformer/converter at the "base of the structure", what impact will this have on these topologies? A graphic illustration of the platform, rather the textual description, showing the electrical component is recommended.

   – The authors have drawn from knowledge in the literature regarding offshore collection network design, as it is analogous to the electrical system required for MRWT's. Both have the purpose of collecting power from a number of turbines and exporting it to a wider electrical network at one single point of connection. The designs proposed in the paper are applicable to both onshore and offshore MRWT's. The effects of an offshore environment on components is an interesting topic, however, this is out of the scope of this project as it will not affect the results of the study. Large wind turbines are satisfactorily protected from the offshore environments so it is assumed that MRWT nacelles are also satisfactorily protected.

   The base of the structure is misleading and an illustration has been included to clarify this (Figure 1) in the revised version of the paper. By base of the structure, the authors refer to a platform between the top of the tower and the bottom of the wind turbine array. This has been included in the design as it is part of the conceptual design outlined in previous studies.

   – Added clarification about the electrical system being analogous to an offshore collection network. Line 145.

   Figure 1 has been adapted to show the physical layout of the system, and a further description of the system has been added at the start of section three to clarify the support structure design. Line 103 to line 112. Instances of "base of the structure" amended to read "platform" - line 148 and line 166.

10. It would good if you could provide recommendations on how to mitigate the noted disadvantages of each cluster topology or otherwise.

   – Thanks for the suggestion; unfortunately, the disadvantages of the cluster topologies are inherent in the design of those topologies. There is no real way to mitigate them without changing the design so they are no longer cluster

topologies. Because of this reason, we provide to the reader with several topologies to help evaluate the pros and cons of each system.

- No action required.

11. Cost, Mass and Loss Estimation: Typo in Line 178, . . .include "the" . . .

    - Corrected
    - Corrected - line 215.

12. Costs estimate has been presented in Table 1 without stating the type of wind turbine for which it applies. Please clarify.

    - Thanks for the observation. In order to clarify, detail has been added in the revised version of the paper to inform the reader that these cost estimates have been made for pitch regulated, variable speed wind turbines.
    - Updated Table 1 (now Table 2) title to "Summary of mass and cost estimations of each component for variable speed, pitch regulated wind turbines". Wind turbine type also included in Table 1.

13. There are some inconsistencies in the use of units. Please check carefully.

    - Table 1 (now Table 2) will be updated so all cost and mass functions use variables instead of units in the formula. The units expressed consistently where possible in the revised manuscript.
    - Table 1 (now Table 2) has been amended to ensure all formula include variables and not units. Units specified in appropriate subsection of section 6 for each component, and are kept consistent where possible.

14. Gearbox rpm: You may need to update your speed regime classification. Typically, 1. Slow-Speed: rated speed less than 400 rpm. 2. Medium Speed: rated speed of 400 rpm to 1200 rpm. 3. High-Speed: rated speed of 1200 rpm or more.

    - The reviewer is correct in mentioning that a 1500 rpm is considered high speed. The original cost relationship used is for medium speed generators – this relationship has been adapted to a relationship for high speed generators in the revised version of the paper. This does not affect the final results as each system uses the same type of generator.
    - The relationship used for cost and mass estimation has been adapted to be accurate for high speed generators instead of medium speed generators. Section 6.1 (Generators) has been rewritten to describe this method. Information in Table 2 has also been updated to reflect the changes made. All mentions of medium speed generators have been changed to high speed generator.

15. Power electronic converter: Although the estimation of converter mass shows a linear scaling relationship for LV - BTB converters, but this relationship is only limited to low power ratings. Any thought how this would affect the topologies performance and its commercial viability?

– With regards to the converter mass scaling, Figure 10 shows that the linear relationship holds well for power ratings up to 6 MW. Higher power ratings still show a linear relationship, but there is slightly higher variance above 6 MW. Higher ratings are unlikely to be required in MRWTs. In view of this, the authors do not believe this would affect the commercial viability of the MRWT.

– No action required

16. Cables (Line 236): Cable damage is a common occurrence in offshore environment; there is need to consider this in the topology design if this study is to be applied of offshore MRWT.

    – Although cable damage is common in offshore wind farms, cable failure rates are typically much lower compared to other components in an offshore wind farm, like generator and converter failures. The topic of reliability analysis for MRWT's is a very interesting and important topic, but detailed work on this subject is out of the scope of this study. The paper does include a qualitative analysis on reliability, which is highlighted in the comparison table presented at the end (Table 4). Cable failures are considered in this at least in a qualitative manner and in terms of how many turbines would be lost because of a cable failure. A recommendation for future work on the topic of MRWT reliability is recommended in the paper as the authors recognise it is an important area for investigation (line 442).

    – No action required

17. Transformers (Line 248): There is high inconsistent use of units in this section. Please crosscheck all units carefully.

    – Mass estimates are based on power rating in kVA, and cost estimates used are based on VA. This has been changed in the revised version of the paper so both units are consistent. Thanks for the observation.

    – the relationship for mass has been amended to be in VA instead of KVA so both mass and cost estimates are consistent with units. Line 311 and Table 2 (previously Table 1)

18. The conclusions should be improved to include summary of key findings and their impact on MRWT. Although a good investigation of the various components of electrical topologies design have been undertaken, the results have not been completely validated in terms of their optimal values and their impact on the overall performance on MRWT. It is suggested that the authors should consider this doing it

    – Thank you for these comments. A summary of key findings has been included in the conclusion of the revised version of the paper. The authors recognise that numerous parameters like voltage level, number of turbines in clusters and different electrical configurations within the topologies should be considered in future work, but is out of the scope of this project. An additional paragraph has been added to the end of section 8 in which the authors make recommendations for future work on these topics. The authors believe that this paper provides insight into the effectiveness of various topology designs, and although not all parameters are fully optimised, the work can form a base for future projects and full optimisation of the system.

    – Conclusion amended to include summary of key findings - line 474 onwards.

        Short paragraph added to end of section 8 to recommend future work - line 469.

[revised manuscript text omitted]